# Aqueous spice extracts as alternative antimycotics to control highly drug resistant extensive biofilm forming clinical isolates of *Candida albicans*

**Bindu Sadanandan**[1]*, **Vaniyamparambath Vijayalakshmi**[1], **Priya Ashrit**[1], **Uddagiri Venkanna Babu**[2], **Lakavalli Mohan Sharath Kumar**[2], **Vasulingam Sampath**[2], **Kalidas Shetty**[3], **Amruta Purushottam Joglekar**[1], **Rashmi Awaknavar**[1]

1 Department of Biotechnology, M S Ramaiah Institute of Technology, Bengaluru, Karnataka, India,
2 Department of Phytochemistry, Research and Development, The Himalaya Drug Company, Bengaluru, Karnataka, India, 3 Department of Plant Sciences, North Dakota State University, Fargo, North Dakota, United States of America

* bindu@msrit.edu, bindu.sadanandan@gmail.com

**Data Availability Statement:** All relevant data are within the manuscript and its Supporting information files.

## Abstract

*Candida albicans* form biofilm by associating with biotic and abiotic surfaces. Biofilm formation by *C. albicans* is relevant and significant as the organisms residing within, gain resistance to conventional antimycotics and are therefore difficult to treat. This study targeted the potential of spice-based antimycotics to control *C. albicans* biofilms. Ten clinical isolates of *C. albicans* along with a standard culture MTCC-3017 (ATCC-90028) were screened for their biofilm-forming ability. *C. albicans* M-207 and *C. albicans* S-470 were identified as high biofilm formers by point inoculation on Trypticase Soy Agar (TSA) medium as they formed a lawn within 16 h and exhibited resistance to fluconazole and caspofungin at 25 mcg and 8 mcg respectively. Aqueous and organic spice extracts were screened for their antimycotic activity against *C. albicans* M-207 and S-470 by agar and disc diffusion and a Zone of Inhibition was observed. Minimal Inhibitory Concentration was determined based on growth absorbance and cell viability measurements. The whole aqueous extract of garlic inhibited biofilms of *C. albicans* M-207, whereas whole aqueous extracts of garlic, clove, and Indian gooseberry were effective in controlling *C. albicans* S-470 biofilm within 12 h of incubation. The presence of allicin, ellagic acid, and gallic acid as dominant compounds in the aqueous extracts of garlic, clove, and Indian gooseberry respectively was determined by High-Performance Thin Layer Chromatography and Liquid Chromatography-Mass Spectrometry. The morphology of *C. albicans* biofilm at different growth periods was also determined through bright field microscopy, phase contrast microscopy, and fluorescence microscopy. The results of this study indicated that the alternate approach in controlling high biofilm-forming, multi-drug resistant clinical isolates of *C. albicans* M-207 and S-470 using whole aqueous extracts of garlic, clove, and Indian gooseberry is a safe, potential, and cost-effective one that can benefit the health care needs with additional effective therapeutics to treat biofilm infections.

**Funding:** This research was funded by DEPARTMENT OF SCIENCE AND TECHNOLOGY (DST), Government of India, grant number SR/FT/LS-124/2012 to Bindu Sadanandan. The funders had no role in the study design, data collection and analysis, decision to publish, or preparation of the manuscript. URL: https://dst.gov.in/.

**Competing interests:** The authors have declared that no competing interests exist.

## Introduction

Many *Candida* species are known to cause infections, but more than 90% of the cases are caused by *Candida albicans* [1]. *C. albicans* is a commensal yeast colonizing several niches in the body such as skin, hair, nail, oral cavity, gastrointestinal tract, and reproductive tract [2–5]. This organism remains asymptomatic with no apparent infection in the body of healthy individuals, unlike the immunocompromised patients hospitalized with serious conditions where it causes candidiasis [6, 7]. The severity of the infection can vary from superficial to invasive candidiasis. The latter may lead to 40% of bloodstream infections in clinical settings [3]. Candidiasis of the mucosae includes the oral and otitic cavities, esophageal, gastrointestinal, and reproductive tracts (vulvovaginitis in females, and balanitis in males). The incidence of *C. albicans* in denture wearers and in health conditions like HIV, acute leukaemia and patients in long term facilities are 50–65%, 95%, 90%, and 65–88% respectively [8].

*C. albicans* studied previously as planktonic cells are now studied in the biofilm form as it is now known to be in its natural state of growth [9]. The virulence of *C. albicans* is attributed to its ability to form a biofilm which is a mixed environment of microbial cells surrounded by an Extra Cellular Matrix (ECM) with distinct properties as compared to planktonic cells [10–13]. The ECM formed around the biofilm acts as a barrier for the entry of antimycotic agents into the cells [14]. Most *Candida* species including *C. albicans* have developed resistance to the azole category of antifungals. However, recalcitrance to echinocandins like caspofungin is rarely found [15]. The resistance of *C. albicans* to these conventional drugs is a growing cause of concern for treatment involving biofilm infections. *C. albicans* grows on medical devices such as intravenous lines, drains, and catheters dwelling within the human body which are further responsible for many clinical manifestations leading to bloodstream infections [1, 16, 17]. These devices also serve as a substrate for their growth and are tough to treat with conventional antimycotics. The removal of such infected devices remains the only option to treat the complications [18]. Hence, additional therapeutics are required for the control of these biofilms within the human body.

New drug molecules from natural products are potential alternatives [19]. Almost 80% of the world's population mainly relies on traditional medicines involving plant-based therapies according to WHO [20, 21]. Oral Candidiasis is one of the common oral mucosal infections and its management is usually very challenging either due to treatment failure or recurrence [22]. In this context, plant extracts more so water extracts with antimycotic properties can be potential alternatives as they are effective and non-toxic to the system [23–26]. Garlic (*Allium sativam* L.), Clove (*Syzygium aromaticum* L.), and Indian Gooseberry (*Phyllanthus emblica* L.) have been explored in traditional medicine for potential therapeutics for conditions such as cancer, constipation, and other disease conditions [27–30]. They have properties such as anti-inflammatory, anti-fungal, anti-bacterial, and antioxidant activities. Allicin, a sulfur compound predominantly present in garlic is one of the active constituents with known antimicrobial activity. Clove consists of eugenols, gallic acid derivatives, and many other phenolic compounds and flavonoids namely ellagic acid and salicylic acid which are responsible for the pharmacological properties [31]. Likewise, Indian gooseberry also possesses important cancer chemopreventive potential and many of the polyphenols such as gallic acid and ellagic acid are found in Indian gooseberry that could be responsible for their potential antimicrobial activity [32, 33]. It is one of the main components present in Triphala which is a concoction of three different ayurvedic medicines [34]. Garlic, gooseberry, and clove extracts have been previously studied for their biocompatibility [35–38]. The cytotoxicity varies with the variety of spices used and the method of preparation of the extract [39]. Garlic has been used to treat and control many diseases like cardiovascular, blood pressure, metabolic disorders, etc [40, 41], and is

also known to improve the symptoms in persons with various fatigues [42]. Gooseberry has proven to improve total cholesterol and triglyceride levels in dyslipidemia patients [43]. When studied in a rat model, a decrease in lipid peroxidation and an increase in glutathione peroxidase and catalase were observed when treated with clove extract [44]. Garlic extract stimulated ROS production which is beneficial for normal cells and fatal for cancer cells [39]. When garlic extract was studied on the KB cancer cell line, a 50% reduction in the cells was observed at a concentration greater than 100 $\mu gmL^{-1}$ [45]. Similar results were observed in the Colo 205 cancer cell line [46]. Alexa et al studied the clove oil and showed concentration-dependent cell cytotoxicity on HaCaT, HGF A375, and SCC-4 cancer cells [47]. Indian gooseberry showed a cytotoxic effect above 100 $\mu gmL^{-1}$ on $HepG_2$ cancer cells [48].

The present study investigates the potential of spice extracts of garlic, clove, and Indian gooseberry in controlling high biofilm-forming multidrug-resistant clinical isolates and standard culture of *C. albicans*. The phytochemical profiling for each of these spice extracts was determined by LC-MS and HPTLC methods. MIC of the spice extracts was determined by broth microdilution and the structural characterization of the cultures was carried out by bright field, phase contrast, and fluorescence microscopic studies. This study is focused on exploring the potential of these spice extracts in controlling drug-resistant *C. albicans* and biofilm formation which can contribute to the potential of new natural therapeutics as antimycotic solutions.

## Materials and methods

### Cultures and identification

*C. albicans* MTCC-3017 (ATCC-90028) was procured from IMTECH (Chandigarh, India). Ten *C. albicans* clinical isolates; M-207 from the umbilical vein catheter of a female baby hospitalized in the Intensive Care Unit (ICU) for a month, M-529 from a vaginal swab of a female patient, S-470 from the sputum of a female patient, U-2647, U-3713, U-3800, U-3893, U-427, U-499 from the urine of female and male patients, D-4 from a dental sample of a male patient, were isolated from patients with invasive candidiasis. All the clinical isolates were isolated, identified, and kindly provided by The Department of Microbiology, M S Ramaiah Medical College and Teaching Hospital, Bengaluru, India. The identity of these isolates was further confirmed at our laboratory by the lactophenol cotton blue test [49], germ tube test [50], and colony morphology on CHROMAgar *Candida* [51].

The formation of biofilm was determined using a biofilm tube test [1].

Ethical clearance was not required for this study as no humans were directly involved.

### Subculturing and maintenance of cultures

All the *C. albicans* isolates were sub-cultured in Malt Yeast Agar (MYA) plates and incubated at 37°C for 24 h. The isolates were stored as glycerol stocks (15%v/v) and maintained at -86°C. All the isolates were propagated in the required medium for further experiments.

### Point inoculation

A loopful of the culture of each isolate was picked using an inoculation loop and inoculated at the center of the Tryptic Soy Agar (TSA) plate. The plates were incubated overnight at 37°C for 16 h.

### *In vitro* induction of *C. albicans* biofilm and the effect of FBS on biofilm adhesion

Biofilm was induced on 96-well microtiter plates with some modifications in the protocol [52]. The cell density of the isolates was standardized to $10^6$ cellsmL$^{-1}$ using a hemocytometer

count. A 100 μL volume of the culture was added to each well of the microtiter plate. The plates were incubated at 37˚C for 90 min to allow the biofilm to adhere to the substratum. After 90 min, the cells were discarded and given a 1X Phosphate Buffered Saline (PBS) wash to remove the unadhered cells. A 100 μL volume of fresh medium was added and incubated for different time points. The growth absorbance was measured at 600 nm using a Biotek Synergy HT microtiter plate reader.

One set of the microtiter plates was pre-coated with Fetal Bovine Serum (FBS) while the other set was not pre-coated with FBS. For the pre-coating, 100 μL of FBS was added to each well and incubated at 37˚C for 24 h, following which the FBS was discarded and the wells were washed with 200 μL of PBS. Biofilm was induced on both sets of the microtiter plates in different media, Malt Yeast Broth (MYB), Yeast Extract Peptone Dextrose (YEPD), Potato Dextrose Broth (PDB), Sabouraud Dextrose Broth (SDB), Yeast Mannitol Broth (YMB), Tryptic Soy Broth (TSB), and RPMI-1640.

## Antimicrobial activity of spice extracts in controlling *Candida* biofilms

**Preparation of aqueous and organic solvent spice extracts.** The raw materials were procured locally and authenticated by The Department of Pharmacognosy, The Himalaya Drug Company, Makali, Bengaluru, India. The extracts were prepared using a previously established protocol [53]. The raw materials were washed thoroughly with tap water followed by a sterile water wash and then air-dried. Aqueous extracts of garlic and Indian gooseberry were prepared by crushing 10 g of the sample in 5 mL of sterile water in a pestle and mortar. An aqueous extract of clove was prepared by mixing 5 g of powdered clove with 10 mL of water. These samples were centrifuged at 10000 rpm for 10 min at 4˚C. The supernatant was filtered using a Whatman filter paper and the whole aqueous extract was used for the study. The dilutions of extracts from 5–200 mgmL$^{-1}$ were prepared for the experiments. The preparation of organic solvent extracts of garlic, clove, and Indian gooseberry involved the same process as followed for the preparation of aqueous extracts however organic solvents such as petroleum ether, ethyl acetate, chloroform, methanol, ethanol, and butanol were used instead of sterile water.

**Agar diffusion assay to determine zone of inhibition.** Agar well and disc diffusion assays were performed [19] for the aqueous and organic solvent extracts to determine the Zone of Inhibition (ZOI). Muller Hinton Agar (MHA) plates were prepared. An inoculum of 10$^6$ cellsmL$^{-1}$ of the culture was spread evenly on the MHA surface. Even-sized wells were punched on the agar plates and sealed with 1% agarose. A 100 μL volume of the spice extracts was added into the wells, sealed, and incubated at 37˚C for 16 h and the ZOI was observed the following day. Sterile distilled water and organic solvents were used as respective controls for aqueous and organic extracts.

Filter paper discs with a 5 mm diameter were supersaturated with the spice extracts and placed on MHA for disc diffusion assay. CLSI standard Fluconazole disc (25 mcg) and Caspofungin acetate (8 mcg) were used to compare the ZOI with aqueous and organic solvent extracts.

**Broth microdilution assay to determine the minimum inhibitory concentration (MIC).** The protocol by [54] was used to determine the MIC by broth microdilution assay. A 100 μL volume of 10$^6$ cellsmL$^{-1}$ was added to microtiter plates and incubated for 90 min at 37˚C in an incubator for adhesion. After 90 min the cells were discarded and washed with 200 μL of 1X PBS. A 100 μL volume of fresh medium was added to the wells. A 50 μL volume of aqueous garlic, clove, and Indian gooseberry extracts with concentrations 0.05–10 mg (dry weight measurements) was used for the treatment, and sterile distilled water was used as the control. The microtiter plates were incubated for different time points from 0–24 h. Optical

density was measured at 600 nm using a microtiter plate reader, based on which MIC50 was determined. Only aqueous extracts were evaluated from this stage onwards due to the superior antimycotic activity after ZOI assays.

*MTT assay*. To determine the viability of *C. albicans* isolates after treatment with aqueous extracts, an MTT assay was performed using the protocol [55]. After incubation in microtiter plates, the unadhered cells were removed by washing with 200 µL of 1X PBS. Following the wash, 50 µL of 5 mgmL$^{-1}$ MTT was added to each well and incubated for 3 h at 37˚C. After incubation, 150 µL of acidified isopropanol was added to each well to dissolve the formazan crystals formed in the wells. The absorbance was measured at 540 nm using a microplate reader.

**Aqueous extracts of garlic, clove, and Indian gooseberry control biofilm induced on glass slide.** The *C. albicans* M-207 and S-470 were grown in a Petri plate containing TSB media, with a glass slide immersed in it and treated with Garlic– 1 mg for *C. albicans* M-207 and Garlic- 1.25 mg, Clove– 0.215 mg and Indian Gooseberry- 0.537 mg for *C. albicans* S-470 at 24 h. Control plates were maintained in parallel. Visual reduction in the biofilm in the Petri plate and glass slide was recorded. The glass slide was further visualized under the microscope to confirm biofilm inhibition.

**High-performance thin layer chromatography of fresh whole aqueous extracts of garlic, clove, and Indian gooseberry.** HPTLC for whole aqueous extracts of garlic, clove, and Indian gooseberry was performed using Alliin, ellagic acid, and gallic acid respectively as standards. Butanol+2 –Isopropanol+ Acetic acid+ Water in the ratio of 3:1:1:1 [56], Toluene + Ethyl acetate + Methanol+ Formic acid in the ratio of 3:3:0.8:0.2 [57] and Toluene + Glacial acetic acid + Ethyl acetate + Formic acid in the ratio of 2:2:4.5:0.5 were used as mobile phases for garlic, clove, and Indian gooseberry respectively. HPTLC of fresh whole aqueous extracts were developed at 254 nm and 366 nm and derivatized by spraying Ninhydrin (1 mgmL$^{-1}$) and ferric chloride (10 mgmL$^{-1}$) to confirm the presence of alliin, ellagic acid, and gallic acid in the fresh extracts of garlic, clove, and Indian gooseberry respectively. A 10x10 cm pre-coated silica gel 60 F$_{254}$ TLC plate was used to develop the chromatogram for HPTLC. A CAMAG HPTLC system equipped with a sample applicator and linomat V, CAMAG TLC visualizer was used to capture the images.

**Liquid chromatography-mass spectrometry of fresh whole aqueous extracts of garlic, clove, and Indian gooseberry.** HPTLC was followed up with LC-MS for further confirmation of allicin, ellagic acid, and gallic acid based on the specific peaks of the standards and their molecular mass at the specified retention time.

A Shimadzu LC-20AD series pump, DUG-20A3 series degasser, and Luna C18 (250 X 4.6 mm, 5 um) phenomenex column were used for the chromatographic run. A 20 µL volume of aqueous extracts of garlic, clove, and Indian gooseberry was injected through SIL-HTC Shimadzu auto-sampler at ambient temperature achieved through CTO-10 AS VP column oven at 30˚C, 40˚C, and 40˚C respectively. The mobile phase was delivered at a flow rate of 1 mLmin$^{-1}$ with a splitter for 25 min for garlic, 38 min for clove, and 49 min for Indian gooseberry.

For the garlic extract, the mobile phase consisted of water (J.T.Baker brand) in pump A and acetonitrile (J.T.Baker brand) in pump B. The flow into the columns was as follows; 0.01 min-5% (pump B), 10 min-20% (pump B), 15 min-40% (pump B), 20 min-40% (pump B), 23 min-5% (pump B) and equilibrium for 2 min-5% (pump B). Peak separation was monitored at 210 nm using SPD-20A UV/VIS detector.

For the clove extract, the mobile phase consisting of water with 0.1% formic acid in pump A and methanol (J.T.Baker brand) in pump B was used. The gradient mobile phase flow into the column was as follows; 0.01 min-15% (pump B), 3 min-20% (pump B), 8 min-35% (pump

B), 18 min-50% (pump B), 28 min-65% (pump B), 32 min-65% (pump B), 35 min-15% (pump B) and equilibrium for 3 min-15% (pump B). Peak separation was monitored at 254 nm and 215 nm.

For the Indian gooseberry extract too the mobile phase consisted of water with 0.1% formic acid in pump A and methanol in pump B. The gradient mobile phase flow into the column was as follows; 0.01 min-0% (pump B), 5 min-0% (pump B), 10 min-3% (pump B), 12 min-5% (pump B), 1 min-10% (pump B), 20 min-20% (pump B), 25 min-35% (pump B), 34 min-50% (pump B), 37 min-65% (pump B), 40 min-90% (pump B), 43 min-90% (pump B), 46 min-0% (pump B) and equilibrium for 3 min-0% (pump B). Peak separation was monitored at 275 nm.

**Bright-field microscopy.** In this study, two staining methods viz. Crystal violet (CV) and lactophenol cotton blue were used to visualize the effect of the spice extracts on *C. albicans* biofilms. *C. albicans* cultures were grown for different time intervals at 24, 48, 72, 96, 120 h, and 12 days in the TSB medium. Based on ZOI and MIC from initial studies, *C. albicans* M-207 treated with aqueous extract of garlic and *C. albicans* S-470 treated with aqueous extracts of garlic, clove, and Indian gooseberry were studied using bright-field microscopy and CV staining at 1, 3 17, 24, 48, 72, 96 and 120 h. Control samples were also observed and compared in parallel.

*Crystal violet staining. Candida* biofilms produced in TSB broth culture were smeared on a glass slide/coverslip and heat fixed. A few drops of 0.1% crystal violet solution prepared in distilled water were added to the glass slide and left for a minute. Excess stain was washed off using distilled water and blotted. The slides/coverslips were observed under a trinocular light microscope at 40X and 100X magnifications. Studies were performed at different time intervals.

*Lactophenol cotton blue staining.* The slides were prepared similarly to that of crystal violet staining. Instead of the crystal violet stain, lactophenol cotton blue was used. The results were observed under a trinocular optical microscope at 40X and 100X magnifications. Studies were performed at different time intervals.

**Phase-contrast microscopy.** Biofilm was induced on microtiter plates as described in the previous protocol. Cultures grown on 96 well plates were directly captured at 1, 3, 6, 12, 24, 48, 72, 96, and 120 h for control and aqueous garlic treated *C. albicans* M-207 and aqueous garlic, clove, and Indian gooseberry treated *C. albicans* S-470. Phase-contrast microscopic images of the biofilm from 1–120 h were captured at 20X magnification (ZEISS, Axio Vert.A1).

**Fluorescence microscopy.** Fluorescence microscopy was carried out for control and treatment samples using calcofluor white dye. Cultures were grown on a glass slide immersed in TSB medium with 1 mg of aqueous garlic extract for *C. albicans* M-207 and 1.25 mg of garlic extract, 0.215 mg of clove extract, and 0.537 mg of Indian gooseberry extract for *C. albicans* S-470 for 3, 6, 12 and 24 h. Control sets for both isolates were maintained in parallel. The glass slide was removed, and the heat fixed. A 200 μL volume of 1 mgmL$^{-1}$ Calcofluor white dye (Fluka analytical, 18909-100ML-F) was added to the glass slide and left for a minute in dark. A 10% KOH solution was added on top of the Calcofluor dye and left for a minute and the excess stain was blotted. The images were captured at 20X magnification under the UV range at an excitation wavelength of 380 nm and emission wavelength of 475 nm using a Fluorescence microscope (ZEISS, Axio Vert.A1).

## Statistical analysis

One-way ANOVA followed by Tukey's multiple comparison tests was used to study the statistical significance between biofilm formation in microtiter plates coated with and without FBS.

## Results and discussion

### Cultures and identification

In this study, we have used a standard *C. albicans* culture along with 10 clinical isolates from several patients with invasive Candidiasis. These clinical isolates are local strains and are either high or low biofilm formers. Multiple identification tests were performed to re-confirm the cultures as *C. albicans*.

*Lactophenol cotton blue*. On staining with lactophenol cotton blue, the cells appear dark blue against a light blue background (S1 Fig) [58]. We can also observe the formation of pseudohyphae in some of the isolates. This is a basic identification test for fungi [59].

*Germ tube test*: We observed the growth of the germ tube in each of the isolates (S2 Fig). Germ tubes are associated with increased synthesis of protein and ribonucleic acid. Amongst all the *Candida* species, only *C. albicans* forms a germ tube. The germ tube and the filament formed are known to be associated with the virulence of the culture [60]. It is also known to induce filamentous growth in *C. albicans* and modulate its growth [61, 62].

*CHROMAgar* Candida. *C. albicans* isolates formed a greenish-blue colony when grown on CHROMAgar Candida medium [51, 63] (S3 Fig). CHROMAgar *Candida* test on a differential agar medium is used to identify different species of *Candida* as per the colour of the colony formed on the agar plate. The chromogenic β-glucosaminidase substrate in the medium is metabolized to give green colonies of *C. albicans* [64].

*Biofilm tube test*. *C. albicans* M-207, *C. albicans* M-529, *C. albicans* S-470, and *C. albicans* U-3800 showed a dark ring of crystal violet stain around the tubes and were categorized as high biofilm formers (S4 Fig). The biofilm tube test is an indicator of the formation of biofilm by the isolates. The thickness and intensity of the ring of crystal violet stain around the tube also to a certain extent indicates whether the isolate is a low, medium, or high biofilm former [65].

### Point inoculation

It was observed that on point inoculation, *C. albicans* M-207 and *C. albicans* S-470 formed a lawn completely covering the Petri plate at 16 h on TSA medium (Fig 1) as compared to the standard culture MTCC-3017 and the other clinical isolates (S5 Fig). Hence, of the 11 isolates, *C. albicans* M-207 and *C. albicans* S-470 were identified as high biofilm formers and chosen for further studies. It was also observed that for the two cultures extensive biofilm was formed in the TSA medium as compared to YEPD (S6 Fig), and SDA (S7 Fig). Therefore, the TSA medium was used for further experiments. The experiment was performed to observe colony

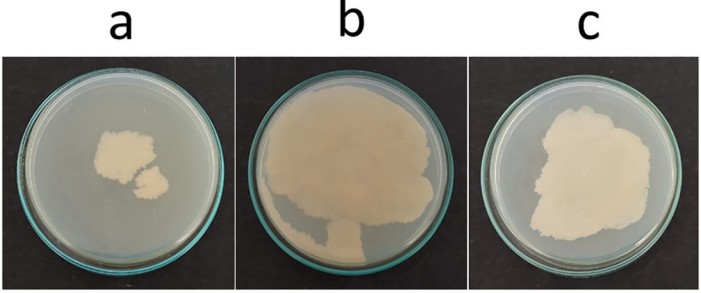

**Fig 1. Point inoculation of *C. albicans* isolates.** (a) MTCC-3017, (b) *C. albicans* M-207, (c) *C. albicans* S-470 on TSA medium at 16 h.

morphology and the ability of the organism to form biofilm on a solid substratum as previous studies have proven that the biofilm-forming capability is highly influenced by the culture medium chosen [10, 66, 67].

### *In vitro* induction of *C. albicans* biofilm and the effect of FBS on biofilm adhesion

Biofilm formation in non-FBS and FBS-coated microtiter plates in different media *viz* MYB (S8 Fig), YEPD (S9 Fig), PDB (S10 Fig), SDB (S11 Fig), YMB (S12 Fig), RPMI-1640 (S13 Fig) and TSB (Fig 2) indicated that FBS did not significantly affect cell adhesion and biofilm formation in *C. albicans* M207 and *C. albicans* S-470. We have previously reported that the FBS coating of microtiter plates did not affect cell adhesion and biofilm formation in some of the clinical isolates of *C. albicans* when grown in RPMI-1640 [68], whereas for the other isolates, FBS did play a significant role. We have also reported in our previous study with *C. glabrata* cultures where biofilm formation in the absence of FBS coating did not make any significant difference to the cell adhesion which is most vital during biofilm formation [69, 70]. FBS has conventionally been used to mimic the *in vivo* environment [71–74]. It is known to provide growth factors, attachment factors, vital nutrients, and toxin scavengers that support the

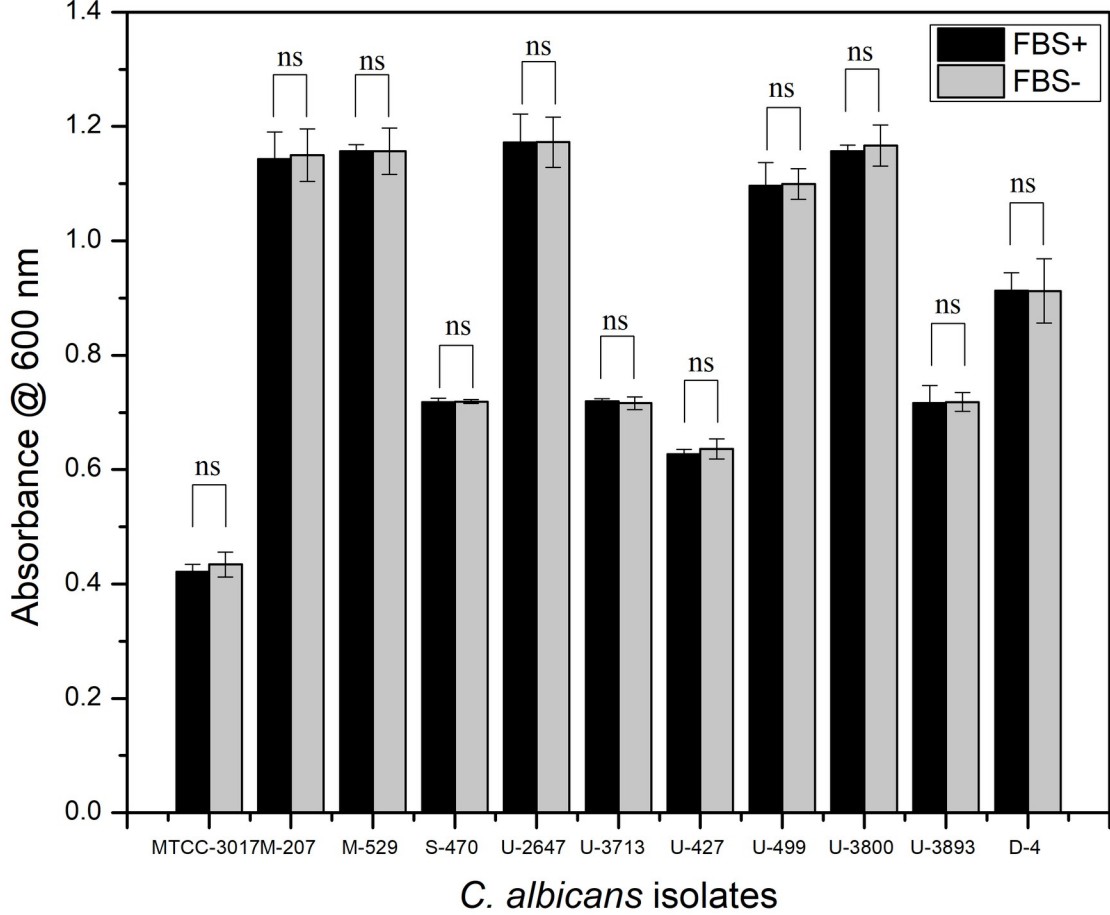

**Fig 2. Growth of *C. albicans* clinical isolates on FBS and non-FBS coated 96 well microtiter plate in TSB medium.** All values are expressed as mean and standard deviation. The experiment was performed in triplicate.

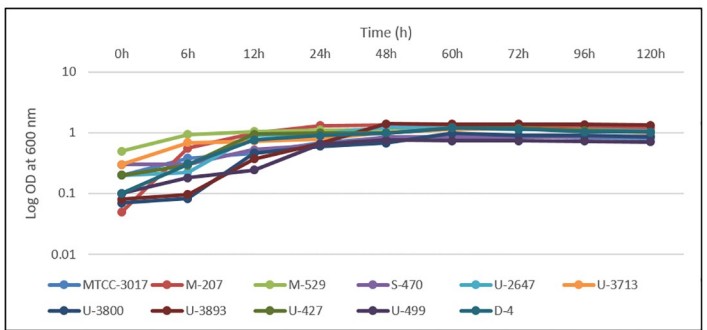

**Fig 3. Growth profile of *C. albicans* isolates in TSB medium for 0–72 h.** All values are expressed as mean and standard deviation. The experiment was performed in triplicate.

growth of biofilms [75]. However, there are also studies proving that FBS inhibits the growth of biofilm in *Staphylococcus aureus* [76]. In this study, strong adherence of *C. albicans* M-207 and S-470 isolates to the substrate material was observed even in the absence of FBS, hence, further studies were carried out without FBS coating.

Growth of *Candida* biofilms in 96 well polystyrene plates was induced in TSB (Fig 3), RPMI-1640 (S14 Fig), and YEPD (S15 Fig). In the YEPD medium, beyond 24 h, the growth rate reduced, whereas, in RPMI-1640 and TSB the cultures grew steadily up to 48 h. The highest growth was observed in the TSB medium and therefore was chosen for further experiments. *C. albicans* is known to grow the best in YEPD, RPMI, TSB, and MYB media [10, 77, 78]. It was also observed that in TSB, the growth of *C. albicans* M-207, S-470, U-2647, U-3893, and U-499 reduced beyond 48 h, whereas in *C. albicans* MTCC-3017, U-3800, U-427, and D-4 it was beyond 60 h and for *C. albicans* M-529, and U-3713 after 72 h. This indicated that the growth of the *C. albicans* cultures is influenced by the choice of media and the culture strain. Previous studies have also shown that the choice of growth media plays a major role in adhesion and biofilm formation *in vitro* [10, 79]. Also, clinical isolates are known to behave differently from standard cultures [80]. This is attributed to several patient-specific factors such as the source of isolation, gender, age, rate of infection, etc.

Adhesion to the substrate material is very important for biofilm formation. The structure, composition of the material, roughness, and hydrophobicity influences the adhesion [20]. Studies have proven that *C. albicans* forms biofilm quickly on the polypropylene material without any requirement of serum [81].

## Antimycotic activity of spice extracts in controlling *Candida albicans* biofilms

**Agar well and disc diffusion.** Various aqueous and organic solvent extracts of Mint (*Mentha arvensis* L.) leaves Turmeric (*Curcuma longa* L.) rhizome, Lemon (*Citrus x limon* L.) fruit, Pepper (*Piper nigrum* L.) corn, Papaya (*Carica papaya* L.) seeds, Papaya leaves (*Carica papaya* L.), Onion (*Allium cepa* L.), Neem (*Azadirachta indica* A. Juss) leaves, Swallow root (*Decalepis hamiltonii*) rhizome (S16 Fig), Garlic (*Allium sativam* L.) cloves, Indian Gooseberry (*Phyllanthus emblica* L.) fruit, and Clove (*Syzygium aromaticum* L.) buds were tested for their antimycotic properties to control *C. albicans* biofilms, amongst which aqueous extracts of garlic, clove, and Indian gooseberry showed most promising results and the ZOI was measured (Fig 4, Table 1). The aqueous extract of lemon also showed a faint inhibition for *C. albicans* M-207.

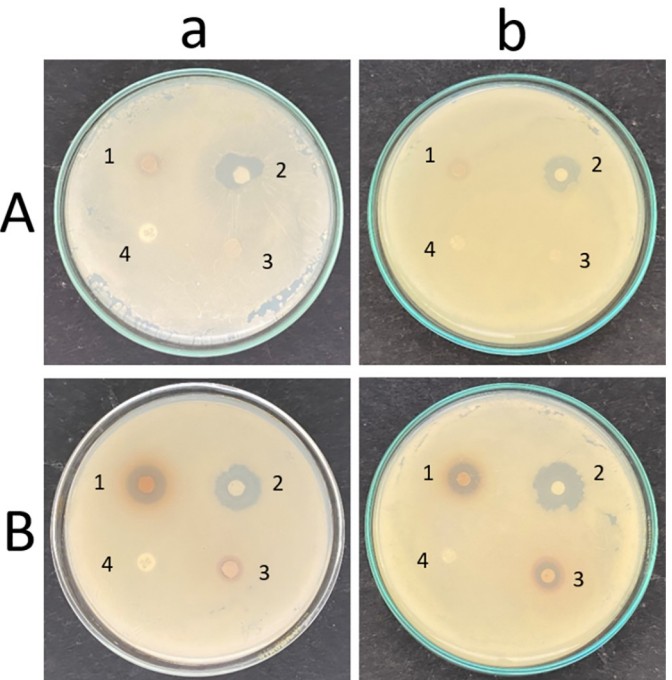

**Fig 4. Antimicrobial activity of aqueous extracts of garlic, clove, and Indian gooseberry with conventional antimycotics.** (1) Clove (43mg), (2) Garlic (200mg), (3) Indian Gooseberry (86mg) with (a) (4) Fluconazole (25mcg) (control) and (b) (4) Caspofungin (8mcg) (control) for (A) *C. albicans* M-207, (B) *C. albicans* S-470. Concentrations are expressed as dry weight measurements.

Garlic and clove are basic spices used the world over. Indian gooseberry is used to prepare spiced pickles and chutneys in India and several other Asian countries. These spices have been used in traditional medicine for ages and are known to have antifungal properties against *Candida* [20, 82, 83]. Garlic has been proven to inhibit the growth of several bacterial species such as *Lactobacillus acidophilus*, *Staphylococcus aureus*, *Pseudomonas aeruginosa*, [84], *Bacillus subtilis*, etc. It has also been proven to inhibit *C. albicans-E. coli* co-cultures [70]. Clove has been used for culinary purposes and produces an essential oil that is used in perfumes, and flavoring and is known for its antimicrobial properties [85]. Indian gooseberry has been used as an anti-inflammatory, diuretic, and cough suppressant and is also known to have an anti-tumor effect [86]. ZOI is a clear zone observed around the region where the cultures do not grow, indicating their antimicrobial activity. It was observed that the chosen *C. albicans*

**Table 1. Zone of inhibition for aqueous extracts of garlic, clove, Indian gooseberry, and standard antimycotic discs of fluconazole and caspofungin by disc diffusion method.**

| Spice extracts/ Conventional Antimycotics | Zone of Inhibition (mm) for *C. albicans* M-207 | Zone of Inhibition (mm) for *C. albicans* S-470 |
|---|---|---|
| Garlic (200 mg dry weight) | 6.5 | 8 |
| Clove (43 mg dry weight) | - | 5 |
| Indian Gooseberry (86 mg dry weight) | - | 4 |
| Fluconazole (25mcg) | - | - |
| Caspofungin (8mcg) | - | - |

cultures were resistant to fluconazole (25mcg) and caspofungin (8mcg) showing no ZOI as observed by the agar disc diffusion method showing no ZOI. The standard fluconazole and caspofungin discs were used as control. The aqueous and some of the organic solvent extracts of garlic, clove, and Indian gooseberry showed ZOI proving the strength of the extracts in controlling the biofilm (Fig 4, Table 1, S17 and S18 Figs, S1 Table). Aqueous extract of garlic was very effective in controlling *C. albicans* M-207 and *C. albicans* S-470, whereas aqueous extracts of clove and Indian gooseberry were effective only against *C. albicans* S-470. The Total Solid (TS) content was estimated for each of the extracts and was found to be 200 mg for garlic, 43 mg for clove, and 86 mg for Indian gooseberry (all dry weight measurements). Further, in the study only the aqueous extracts were considered in the study as they were more efficient in controlling *C. albicans* based on ZOI and also did not have any toxicity issues as opposed to the organic solvent extracts.

Another unique observation was that for *C. albicans* M-207 and *C. albicans* S-470 when treated with a low concentration of garlic extract, the morphology of the biofilm appeared to be creased. At higher concentrations, a clear zone of inhibition was observed around the wells and the biofilm formed away from the wells does not have the creased morphology and instead seems to be smooth (Fig 5).

**Broth microdilution to determine minimum inhibitory concentration (MIC).** The results showed that at 12 h a 50% reduction in growth was observed for *C. albicans* M-207 as measured by growth absorbance at 600 nm with concentrations of 1 mg garlic and 1.25 mg garlic, 0.215 mg clove, and 0.537 mg Indian gooseberry for *C. albicans* S-470 (Fig 6). The cultures were treated with different concentrations of garlic (0.25 mg-10 mg), clove (0.05 mg-0.537 mg), and Indian gooseberry (0.107 mg-4.3 mg), for 0–24 h. The least optical density measured at the least concentration of extracts for which there is a 50% reduction of cells is

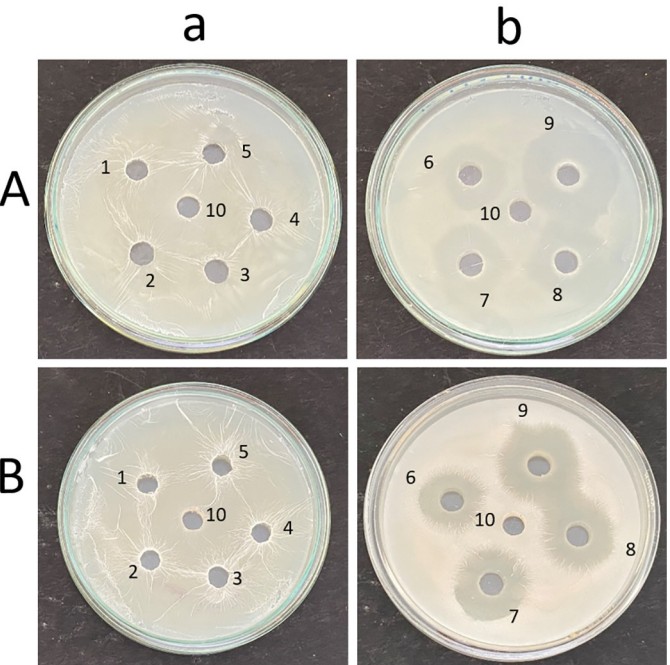

**Fig 5. Antimicrobial activity of aqueous garlic extract.** (a) garlic extract at 0.25mg—1.25mg (b) garlic extract at 2.5 mg– 200 mg (Neat) on MHA for (A) *C. albicans* M-207 & (B) *C. albicans* S-470. Concentrations expressed as dry weight measurements.

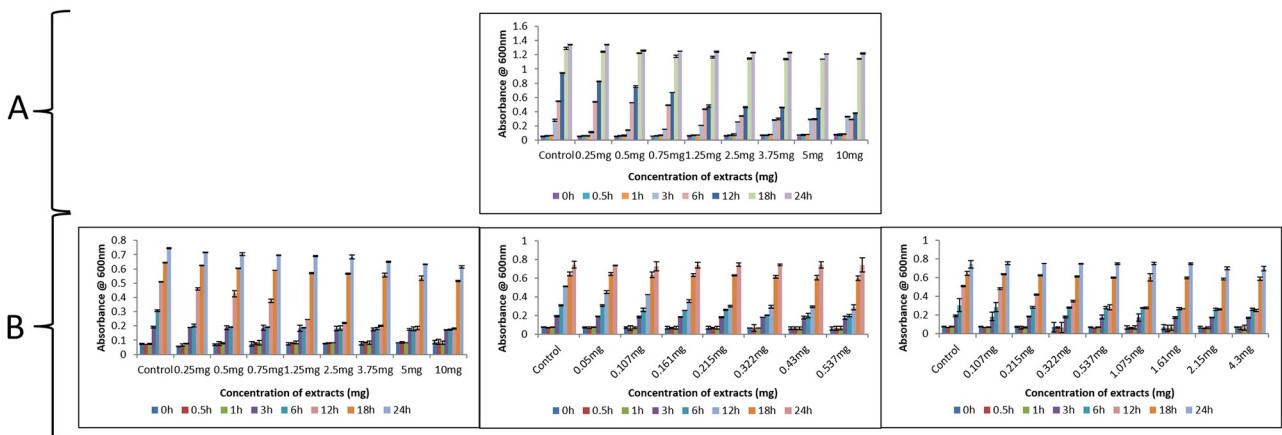

**Fig 6. Antimicrobial activity of aqueous extracts of garlic, clove, and Indian gooseberry on 96 well microtiter plate.** (A) *C. albicans* M-207, (B) *C. albicans* S-470 at various concentrations and different time intervals by growth absorbance. All values are expressed as mean and standard deviation. The experiment was performed in triplicate. Concentrations are expressed as dry weight measurements.

considered to be the MIC [87]. Minimum inhibitory concentration 50 ($MIC_{50}$) is the most widely used and informative measure of a drug's efficacy.

*MTT assay*. MTT assay was performed to determine the viability of cells at $MIC_{50}$. However, we have also carried out MTT at MIC concentrations lower and higher than $MIC_{50}$ (Fig 7). MTT assay was used to quantify the metabolically active cells [88]. At $MIC_{50}$, cell viability of 24.4% was observed for *C. albicans* M-207 when treated with garlic. Similarly, the viability of 26.9%, 29.9%, and 36.1% were observed when *C. albicans* S-470 was treated with garlic, clove, and Indian gooseberry respectively. MTT was also performed for 24 h and 48 h

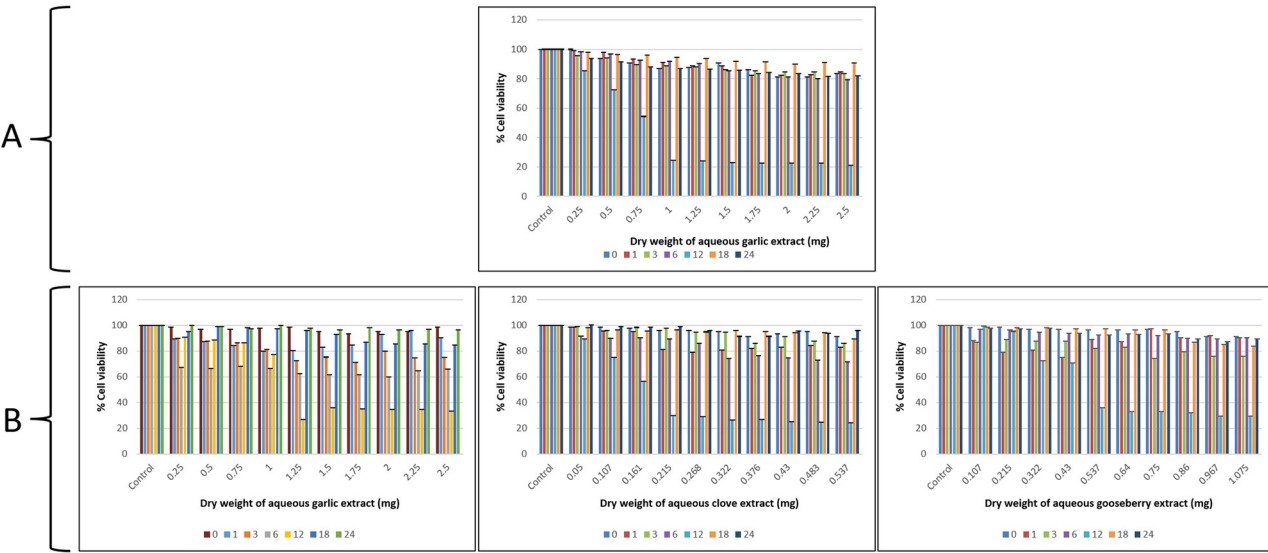

**Fig 7. Panel A: Antimicrobial activity of aqueous extracts of garlic, clove, and Indian gooseberry by MTT Assay.** (A) *C. albicans* M-207, (B) *C. albicans* S-470 at different concentrations and at different time intervals. The values are expressed as mean and standard deviation. The assay was performed in triplicate. Concentrations are expressed as dry weight measurements. **Panel B: Antimicrobial activity of aqueous extracts of garlic against preformed *C. albicans* biofilm by MTT Assay.** (A) 24 h preformed biofilm of *C. albicans* M-207, (B) 24 h preformed biofilm of *C. albicans* S-470, (C) 48 h preformed biofilm of *C. albicans* M-207, (D) 48 h preformed biofilm of *C. albicans* S-470.

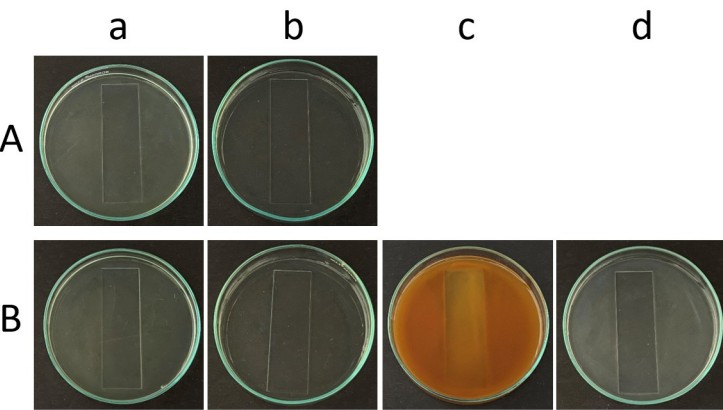

**Fig 8. Glass slides immersed in glass Petri plate containing *C. albicans* treated with garlic, clove, and Indian gooseberry.** (a) *C. albicans* Control, (b) Aqueous garlic treated, (c) Aqueous clove treated, (d) Aqueous Indian gooseberry treated (A) *C. albicans* M-207 and (B) *C. albicans* S-470.

preformed biofilms of *C. albicans* M-207 and S-470 treated with garlic extract at different concentrations and incubation times. For 24 h preformed biofilm, the cell viability at $MIC_{50}$ was 49.27% and 53.1% for *C. albicans* M-207 and S-470 respectively. For 48 h preformed biofilm, the cell viability was 50.49% and 56.57% for *C. albicans* M-207 and S-470 respectively. The percentage of cell viability was observed to be higher in the preformed biofilms as compared to the developing biofilms.

**Aqueous extracts of garlic, clove, and Indian gooseberry control biofilm induced on glass.** The decrease in the biofilm in the aqueous extract-treated glass Petri plates as compared to the control Petri plates can be observed not just under the microscope, but also visually by just looking at the glass microscopic slides in the Petri plates. A reduction in the turbidity can be observed in the Petri plates and glass slides treated with garlic extract for both *C. albicans* M-207 and *C. albicans* S-470 when compared with the control. The clove extract-treated Petri plate looks to be more turbid than the control, mostly due to the color of the clove extract. However, on careful observation, the edges of the slide appear to be clearer in comparison to the control (Fig 8).

**Chemical profiling of aqueous spice extracts.** The presence of a specific band at Rf: 0.38, Rf: 0.3, and Rf: 0.82 were observed adjacent to the standard markers of alliin, ellagic acid, and gallic acid in the track loaded with garlic, clove, and Indian gooseberry respectively that confirm the presence of the desired chemical compound in the fresh aqueous extracts used for the antimicrobial studies (Fig 9). Although ellagic and gallic acid are both present in clove and Indian gooseberry [89, 90], we considered ellagic acid as the standard for clove and gallic acid as the standard for Indian gooseberry, as the stability of ellagic acid is better in clove than in Indian gooseberry and also the gallic acid is present in more quantity in Indian gooseberry than in clove [91, 92]. TLC was observed under 254 nm and 366 nm (S19 Fig) and developed thereafter using spraying reagents.

Conventional methods of extraction mostly involve the use of organic solvents like methanol, ethanol, acetone, etc, to understand and evaluate the pharmacological efficacy of the extracts. However, in our study, aqueous extracts were found to be more effective. Also, they are eco-friendly with no side effects as no organic solvents are used during the extraction. The process is also economical, and reproducible, requiring less process time with efficacy even at low concentrations of the crude aqueous extract.

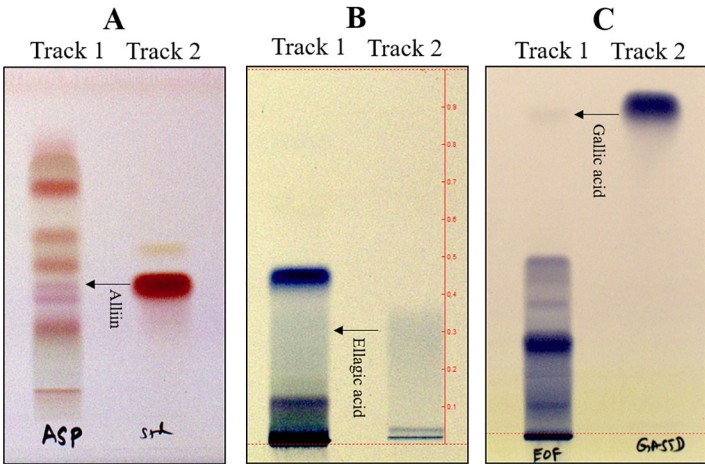

**Fig 9. HPTLC results of garlic, clove, and Indian gooseberry.** (A) Garlic extract derivatized with ninhydrin—Track 1: Garlic (Aq. Extract); Track 2: Alliin standard (Rf: 0.38), (B) Clove extract derivatized with ferric chloride—Track 1: Clove (Aq. Extract); Track 2: Ellagic acid standard (Rf: 0.3), (C) Indian Gooseberry extract derivatized with ferric chloride—Track 1: Indian Gooseberry (Aq. Extract); Track 2: Gallic acid standard (Rf: 0.82).

The results of LCMS showed the presence of peaks for allicin, ellagic acid, and gallic acid for aqueous extracts of garlic, clove, and Indian gooseberry respectively. LC-MS was also performed to further confirm the presence of allicin, ellagic acid, and gallic acid based on the specific peaks observed for these phytochemicals and based on the molecular mass of these components at their specified retention time.

LC-MS for fresh aqueous garlic extract was performed and a peak was obtained at an RT of 19.79 min at 210 nm. The mass spectrometry at the same RT showed the presence of allicin that has undergone protonation (m+h), becoming a molecular ion with a molecular mass of 163.1 mz$^{-1}$ as shown in Fig 10. Allicin, one of the potent compounds present only in garlic was used as a standard marker with a molecular weight of 162.27 gmL$^{-1}$. When garlic is crushed, allicin, an organo-sulfur compound is formed from the precursor alliin in the presence of alliinase.

LC-MS of standard ellagic acid as well as the fresh aqueous clove extract revealed that the peak for the ellagic acid standard was observed at an RT of 23.44 at 254 nm with a molecular mass of 300.9 gmol$^{-1}$. The LC-MS spectra for fresh aqueous clove extract at an RT of 23.54 at 254 nm showed the presence of ellagic acid with a molecular mass of 300.9 mz$^{-1}$ indicating that the compound had undergone deprotonation (m-h), thus becoming a molecular ion (mz$^{-1}$) as shown in Fig 10. Ellagic acid, with a molecular weight of 302.19 gmL$^{-1}$ present in clove was used as the standard marker.

LC-MS of standard gallic acid was performed along with fresh aqueous Indian gooseberry extract at 275 nm. The peak for the gallic acid standard was observed at an RT of 18.69 with a molecular mass of the compound weighing 168.8 gmol$^{-1}$. The LC-MS spectra for fresh aqueous Indian gooseberry extract at an RT of 18.73 showed the presence of gallic acid with a molecular mass of 168.6 mz$^{-1}$ indicating that the compound had undergone deprotonation (m-h), thus becoming a molecular ion (mz$^{-1}$) as shown in Fig 10. Gallic acid, present in Indian gooseberry with a molecular weight of 170.12gmL$^{-1}$ was used as the standard marker.

**Bright-field microscopy.** Bright-field microscopy with crystal violet staining, one of the simplest optical microscopic techniques was used to observe growth and biofilm formation in *C. albicans* M207 and *C. albicans* S470 for up to 12 days without any spice extract

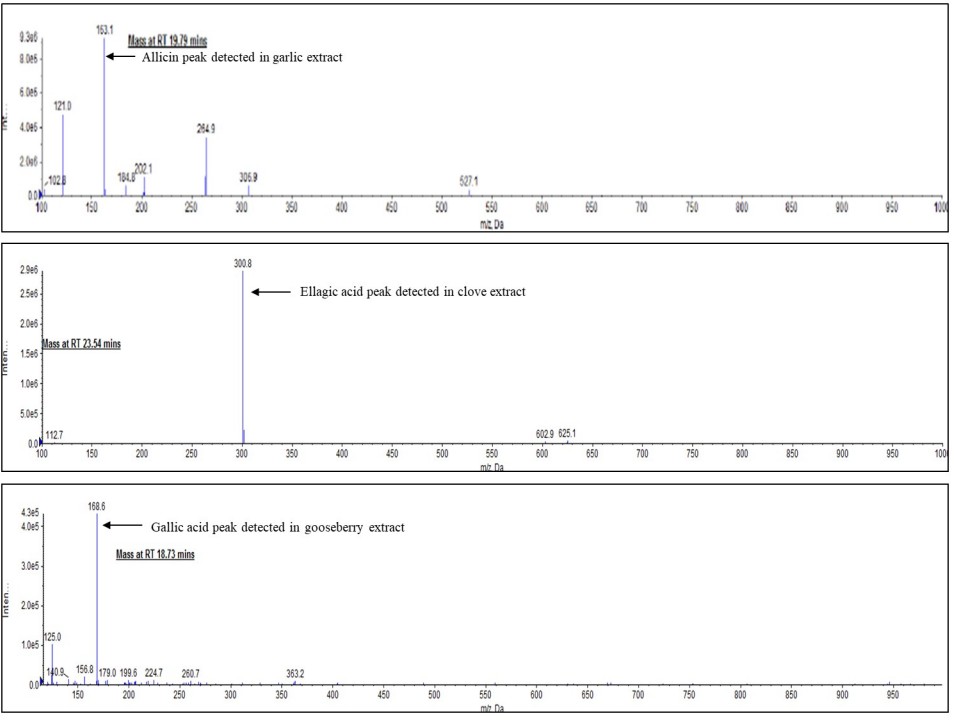

**Fig 10. LC-MS spectra for crude fresh aqueous extract of garlic, clove, and Indian gooseberry.** LC-MS measured at 210 nm, 275 nm, and 254 nm respectively.

treatment. An increase in pseudo hyphae and chlamydospores development with an increase in the incubation time of the cultures was observed (S20 Fig). Similar results were observed for bright field microscopy with lactophenol cotton blue staining as well (S21 Fig). Observation of cells at different stages of growth has been very important to understand any organism [93]. On treatment with aqueous spic extracts, there was an effective decrease in the number of cells in the treatment samples when compared to the control on 12 h of incubation (Fig 11). However, with an increase in incubation time, cell clusters forming pseudo hyphae, hyphae, and biofilm were observed. In a study with clove and lemongrass extracts a reduction in cells was observed at 48 h [32].

**Phase-contrast microscopy.** The control samples were denser with more biomass, whereas, in the treatment samples, a reduction in biomass was observed. Due to the dense biofilm, individual cells could not be observed in phase-contrast microscopy (Fig 12). However, it has the advantage that the cultures need not be stained as in bright field microscopy. This microscopic technique has been useful to study biofilm at different angles [94–96]. However, [97] experienced that the phase-contrast image was not clear for their study. Hence to understand the cells better, higher magnification microscopy such as fluorescence microscopy was performed.

**Fluorescence microscopy.** An increase in the fluorescence in the control sets of *C. albicans* M-207 and *C. albicans* S-470 was observed, with increased incubation periods of 3 h to 24 h as the calcofluor dye binds to the cell walls of *C. albicans*. The bright blue fluorescence background indicates that there was dense biofilm formed as observed in Fig 13, Panel A. Calcofluor white is a UV-excitable dye that contains polysaccharides with β1–3 and β1–4 linkages which bind to chitin and beta-glucan. It has been used extensively to stain fungal cell walls [52,

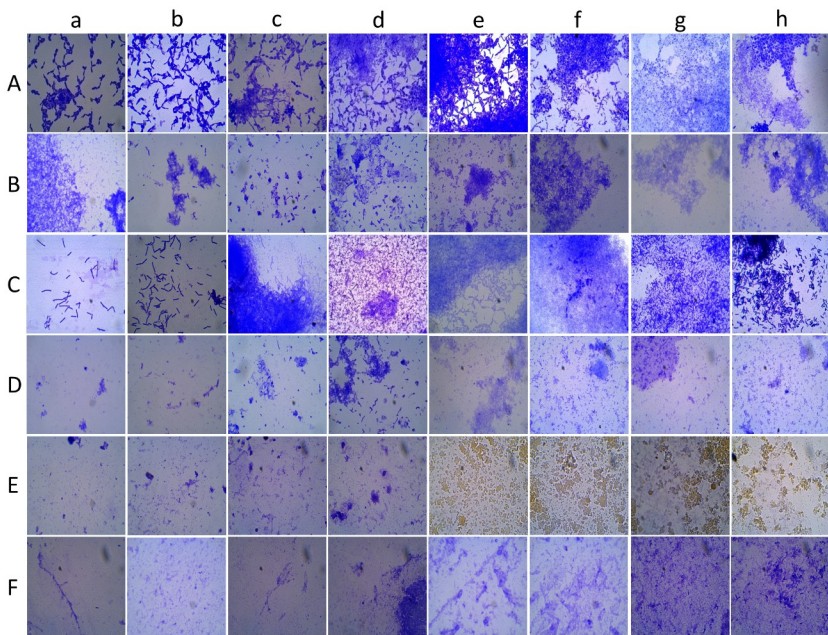

**Fig 11. Microscopic images of crystal violet stained *Candida* biofilm on a glass slide.** Images captured at 100X magnification at different time intervals (a) 1 h, (b) 3 h, (c) 12 h, (d) 24 h, (e) 48 h, (f) 72 h, (g) 96 h (h) 120 h for (A) *C. albicans* M-207 control, (B) Aqueous garlic treated *C. albicans* M-207, (C) *C. albicans* S-470 control, (D) Aqueous garlic treated *C. albicans* S-470, (E) Aqueous clove treated *C. albicans* S-470, (F) Aqueous Indian gooseberry treated *C. albicans* S-470.

98]. At 6 h, the fluorescence begins to reduce indicating the reduction in the number of cells in the treated samples, however, there was a significant decrease in the number of cells at 12 h indicating that aqueous garlic extract was very effective in inhibiting *C. albicans* M-207. This was also confirmed by quantifying the fluorescent images using ImageJ software (Fig 13, Panel B). The intensity of the treated samples was less when compared to the control at all-time intervals, the most effective at 12 h of incubation. Aqueous extracts of garlic, clove, and Indian gooseberry were found to be effective in inhibiting *C. albicans* S-470. Amongst the three extracts, the aqueous extract of garlic was found to be most effective as compared to the aqueous extracts of clove and Indian gooseberry in inhibiting *C. albicans* S-470. Quantification of the images using ImageJ also showed the effectiveness of garlic when compared to other spice extracts on *C. albicans* S-470. At 12 h of incubation, garlic extract had the highest inhibitory effect against *C. albicans* S-470. However, the cells formed biofilm with an increase in ECM that can be seen as an increase in blue fluorescence at 24 h in the treated set probably suggesting that the efficacy of the extract could have been reduced. The increase in the ECM after 24 h can also be attributed to the persistent cells. These persistent cells are known to have higher resistance, survive any antimycotic agents and interestingly are sensitive to growth inhibition and produce new persistent cells [70, 99].

## Conclusions

Controlling multidrug resistant extensive biofilm forming pathogens is a challenge in clinical settings. In light of this, alternative methods to control these pathogens is an essential need. In this study a preventive and prognostic approach in controlling *C. albicans* biofilm, with spice extracts was undertaken. We have reported for the first time a simple and cost-effective

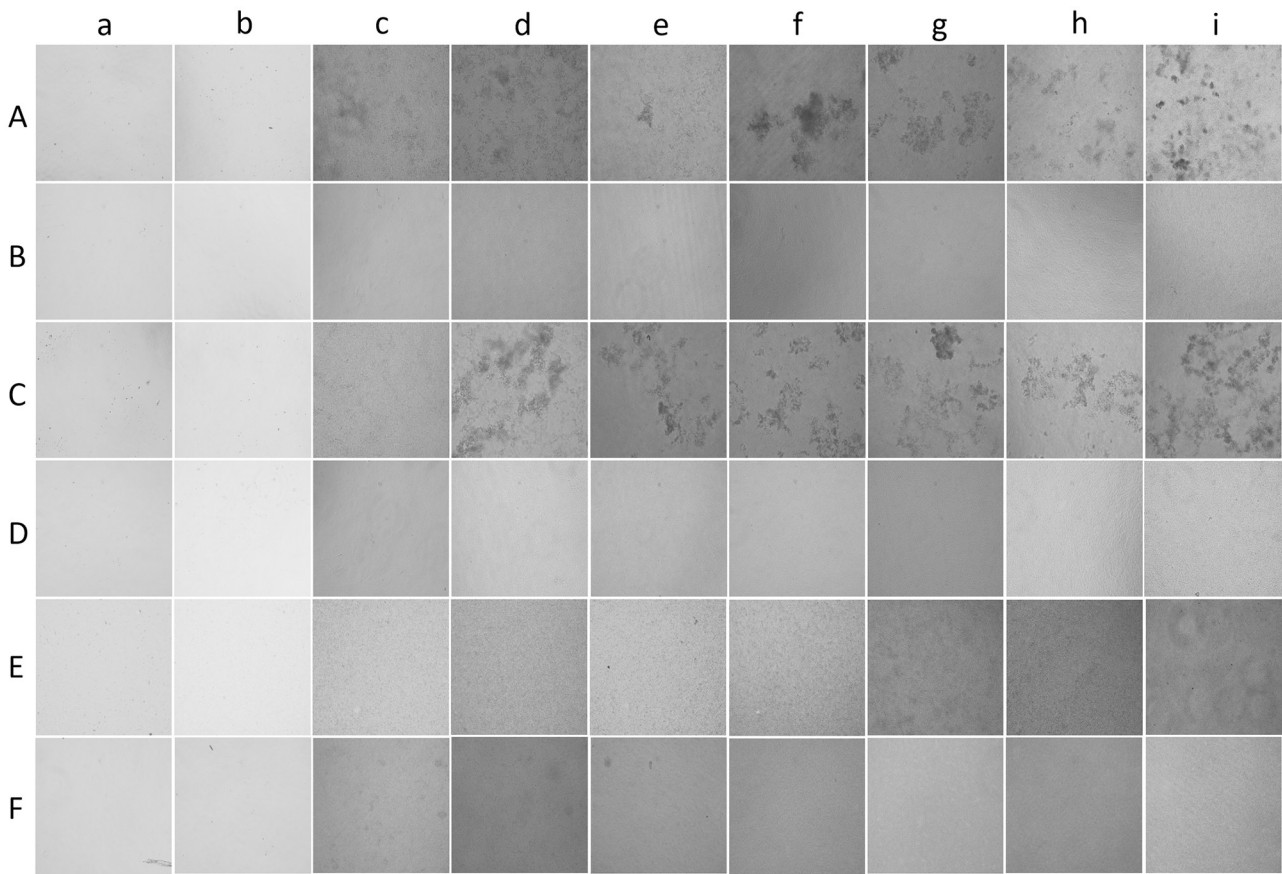

**Fig 12. Phase-contrast microscopic images in 96 well microtiter plate.** Images captured at 20x magnification at different time intervals (a) 1 h, (b) 3 h, (c) 6 h, (d) 12 h, (e) 24 h, (f) 48 h, (g) 72 h (h) 96 h (i) 120 h for (A) *C. albicans* M-207 control, (B) Aqueous garlic treated *C. albicans* M-207 (1mg), (C) *C. albicans* S-470 control, (D) Aqueous garlic treated *C. albicans* S-470 (1.25mg), (E) Aqueous clove treated *C. albicans* S-470 (0.215mg), (F) Aqueous Indian gooseberry treated *C. albicans* S-470 (0.537mg). Concentrations are expressed as dry weight measurements.

method of whole aqueous extracts of garlic, clove and Indian gooseberry to control and inhibit *C. albicans* biofilm formation. The results of the study have revealed that FBS does not have a pertinent effect on cell adhesion and biofilm formation in high biofilm forming MDR strains of *C. albicans*, but the choice of culture media has a major influence on *in vitro* biofilm formation. Aqueous extracts of garlic, clove, and Indian gooseberry were effective in controlling biofilm formation with a minimum concentration of 1 mg garlic for *C. albicans* M-207 and 1.25 mg garlic, 0.215 mg clove and 0.537 mg Indian gooseberry for *C. albicans* S-470 at 12 h of incubation. Chemical profile analysis determined the presence of allicin, ellagic acid, and gallic acid in the aqueous extracts of garlic, clove, and Indian gooseberry respectively. Biochemical and structural characterization studies have demonstrated the reduction in cell numbers on treatment with aqueous spice extracts and confirmed the transition of cells from the adherent biofilm phenotype to sessile non biofilm type. These results are indicative of the effectiveness of the extracts even in their whole forms which could be potentially far more effective when purified. Thus, there is a high scope to translate the knowledge and outcomes from this study into developing novel therapeutic approaches to tackle the threat of *C. albicans* infection. This approach can also be further applied in developing other spice-based antimycotic therapeutics in the form of tablets/injections/powders/sprays/douches and catheters coated with spice extracts such as these targeted long-used spice extracts.

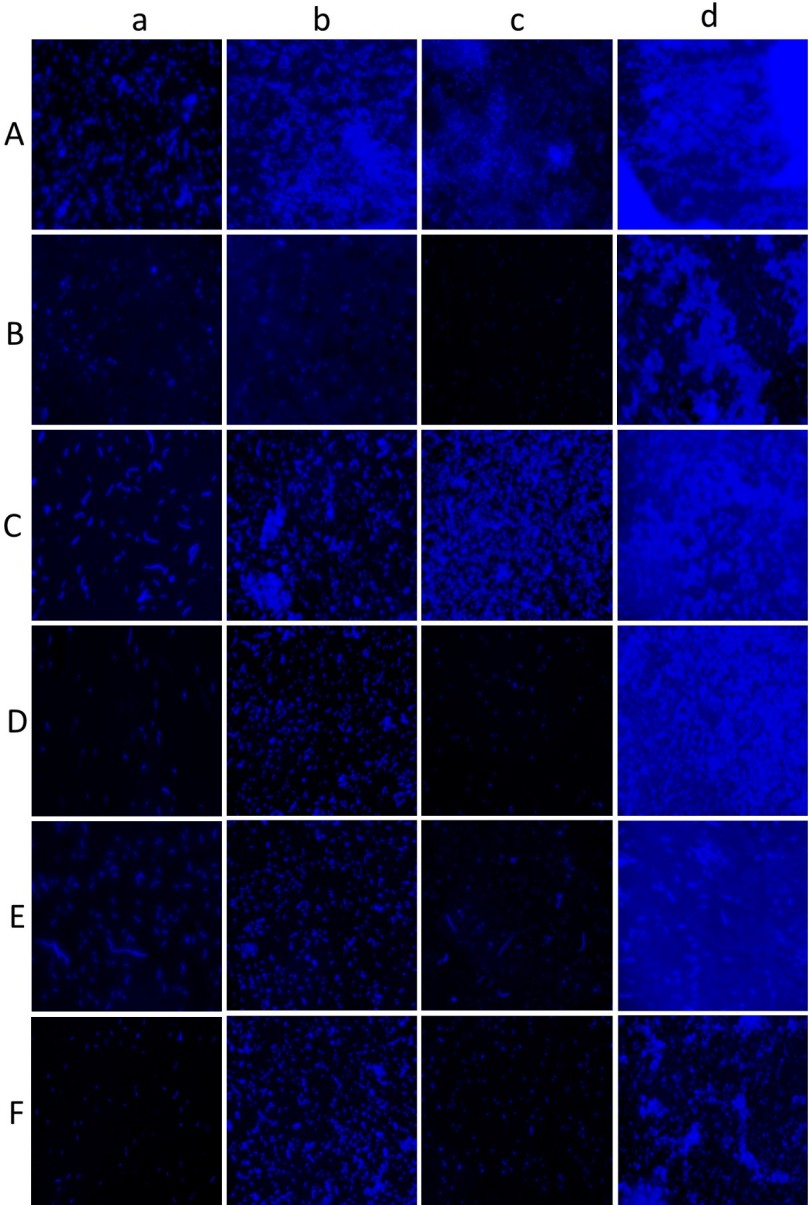

**Fig 13. Fluorescence images of *C. albicans* stained with calcofluor white.** Panel A: Images captured at different time intervals (a) 3 h, (b) 6 h, (c) 12 h, (d) 24 h for (A) *C. albicans* M-207 control, (B) Aqueous garlic treated *C. albicans* M-207 (1mg), (C) *C. albicans* S-470 control, (D) Aqueous garlic treated *C. albicans* S-470 (1mg), (E) Aqueous clove treated *C. albicans* S-470 (0.215mg), (F) Aqueous Indian gooseberry treated *C. albicans* S-470 (0.537mg). All images were captured at 40x magnification. Concentrations are expressed as dry weight measurements. Panel B: Quantification of fluorescence intensity of *C. albicans* M-207 and S-470 treated with garlic, clove, and gooseberry at different time intervals analyzed by the fluorescence of the entire image using ImageJ.

## Supporting information

**S1 Fig. Lactophenol cotton blue stained *C. albicans*.** (a) MTCC-3017, (b) U-427, (c) U-499, (d) U-3893, (e) M-207, (f) M-529, (g) S-470, (h) U-2647, (i) U-3800, (j) U-3713, (k) D4 at 24 h. (TIF)

**S2 Fig. Germ tube formation in *C. albicans* isolates.** (a) MTCC-3017, (b) M-207, (c) M-529, (d) S-470, (e) U-2647, (f) U-3713, (g) U-3800, (h) U-3893, (i) U-427, (j) U-499, and (k) D-4. The arrows indicate the germ tube formed by the isolates.
(TIF)

**S3 Fig. *C. albicans* isolates grown on CHROMagar medium.** MTCC-3017, M-207, M-529, S-470, U-2647, U-3713, U-3800, U-3893, U-427, U-499, and D-4 were cultured.
(TIF)

**S4 Fig. Biofilm tube test for *C. albicans* isolates.** MTCC-3017, M-207, M-529, S-470, U-2647, U-3713, U-3800, U-3893, U-427, U-499, and D-4.
(TIF)

**S5 Fig. Point inoculation of *C. albicans* isolates on TSA medium.** (a) U-427, (b) U-499, (c) U-2647, (d) U-3800, (e) U-3893, (f) U-3713, (g) M-529, and (h) D-4 at 16 h.
(TIF)

**S6 Fig. Point inoculation of *C. albicans* isolates on YEPD medium.** MTCC-3017, M-207, M-529, S-470, U-2647, U-3713, U-3800, U-3893, U-427, U-499, and D-4 at 16 h.
(TIF)

**S7 Fig. Point inoculation of *C. albicans* isolates on SDA medium.** MTCC-3017, M-207, M-529, S-470, U-2647, U-3713, U-3800, U-3893, U-427, U-499, and D-4 at 16 h.
(TIF)

**S8 Fig. Growth of *C. albicans* clinical isolates on FBS and non FBS coated 96 well microtiter plate in MYB medium.** All values are expressed as mean and standard deviation. The experiment was performed in triplicate.
(TIF)

**S9 Fig. Growth of *C. albicans* clinical isolates on FBS and non FBS coated 96 well microtiter plate in YEPD medium.** All values are expressed as mean and standard deviation. The experiment was performed in triplicate.
(TIF)

**S10 Fig. Growth of *C. albicans* clinical isolates on FBS and non FBS coated 96 well microtiter plate in PDB medium.** All values are expressed as mean and standard deviation. The experiment was performed in triplicate.
(TIF)

**S11 Fig. Growth of *C. albicans* clinical isolates on FBS and non FBS coated 96 well microtiter plate in SDB medium.** All values are expressed as mean and standard deviation. The experiment was performed in triplicate.
(TIF)

**S12 Fig. Growth of *C. albicans* clinical isolates on FBS and non FBS coated 96 well microtiter plate in YMB medium.** All values are expressed as mean and standard deviation. The experiment was performed in triplicate.
(TIF)

**S13 Fig. Growth of *C. albicans* clinical isolates on FBS and non FBS coated 96 well microtiter plate in RPMI-1640 medium.** All values are expressed as mean and standard deviation. The experiment was performed in triplicate.
(TIF)

**S14 Fig. Growth profile of *C. albicans* isolates in RPMI-1640 for 0-72h.** All values are expressed as mean and standard deviation. The experiment was performed in triplicate. (TIF)

**S15 Fig. Growth profile of *C. albicans* isolates in Yeast Extract Peptone Dextrose for 0-72h.** All values are expressed as mean and standard deviation. The experiment was performed in triplicate. (TIF)

**S16 Fig. Screening of spice extracts.** (a) (1) Clove (2) Papaya seeds (3) Indian Gooseberry (4) Onion (5) Water as control, (b) (1) Pudina (2) Pepper (3) Lemon (4) Garlic (5) Water as control, (c) Papaya leaf (1) 100 mgmL$^{-1}$ (2) 250 mgmL$^{-1}$ (3) 500 mgmL$^{-1}$ (4) 750 mgmL$^{-1}$ (5) 1000 mgmL$^{-1}$ (6) 2000 mgmL$^{-1}$ (7) Water as control, (d) Neem (1) 100 mgmL$^{-1}$ (2) 250 mgmL$^{-1}$ (3) 500 mgmL$^{-1}$ (4) 750 mgmL$^{-1}$ (5) 1000 mgmL$^{-1}$ (6) 2000 mgmL$^{-1}$ (7) Water as control, (e) Turmeric (1) 100 mgmL$^{-1}$ (2) 250 mgmL$^{-1}$ (3) 500 mgmL$^{-1}$ (4) 750 mgmL$^{-1}$ (5) 1000 mgmL$^{-1}$ (6) 2000 mgmL$^{-1}$ (7) Water as control, (f) Swallow root (1) 100 mgmL$^{-1}$ (2) 150 mgmL$^{-1}$ (3) 250 mgmL$^{-1}$ (4) 500 mgmL$^{-1}$ (5) Water as control. (A) *C. albicans* M-207 and (B) *C. albicans* S-470 biofilm. (TIF)

**S17 Fig. Zones of inhibition (mm) in well diffusion showing antimicrobial activity of garlic, clove and Indian gooseberry on MHA.** (A) *C. albicans* M-207 and (B) *C. albicans* S-470 by solvent extraction. (a) (1) Clove + Petroleum ether, (2) Clove + Ethyl Acetate, (3) Garlic + Petroleum ether, (4) Garlic + Ethyl Acetate, (5) Indian Gooseberry + Ethyl Acetate, (6) Indian Gooseberry + Petroleum ether, (7) Control (Water), (b) (1) Garlic + Chloroform, (2) Indian Gooseberry + Chloroform, (3) Clove + Chloroform, (4) Garlic + Methanol, (5) Indian Gooseberry + Methanol, (6) Clove + Methanol, (7) Control (Water) (c) (1) Garlic + Ethanol, (2) Indian Gooseberry + Ethanol, (3) Clove + Ethanol, (4) Garlic + Butanol, (5) Indian Gooseberry + Butanol, (6) Clove + Butanol, (7) Control (Water). (TIF)

**S18 Fig. Zones of inhibition (mm) in disc diffusion showing antimicrobial activity of garlic, clove and Indian gooseberry on MHA.** (A) *C. albicans* M-207 and (B) *C. albicans* S-470 by solvent extraction. (a) (1) Clove + Petroleum ether, (2) Clove + Ethyl Acetate, (3) Garlic + Petroleum ether, (4) Garlic + Ethyl Acetate, (5) Indian Gooseberry + Ethyl Acetate, (6) Indian Gooseberry + Petroleum ether, (7) Control (Water), (b) (1) Garlic + Chloroform, (2) Indian Gooseberry + Chloroform, (3) Clove + Chloroform, (4) Garlic + Methanol, (5) Indian Gooseberry + Methanol, (6) Clove + Methanol, (7) Control (Water (c) (1) Garlic + Ethanol, (2) Indian Gooseberry + Ethanol, (3) Clove + Ethanol,(4) Garlic + Butanol, (5) Indian Gooseberry + Butanol, (6) Clove + Butanol, (7) Control (Water). (TIF)

**S19 Fig. HPTLC of aqueous extracts of garlic, clove and Indian gooseberry.** (A) HPTLC of Aqueous garlic extract indicated as ASP (*Allium sativum* Pulp)—Track 1: ASP (Aq. extract); Track 2: Alliin standard, (B) HPTLC of Aqueous clove extract indicated as SAFB (*Syzygium aromaticum* Flower Bud)—Track 1: SAFB; Track 2: Ellagic acid standard, (C) HPTLC of Aqueous gooseberry extract indicated as EOF (*Emblica officinalis* Fruit)—Track 1: EOF; Track 2: Gallic acid standard, at (a) 254 nm and (b) 366 nm. (TIF)

**S20 Fig. Microscopic images of crystal violet stained *C. albicans*.** (A) *C. albicans* M-207, (B) *C. albicans* S-470 induced on coverslip in TSB medium for (a) 24h, (b) 48h, (c) 72h, (d) 96h,

(e) 120h, (f) 12days at 40X & 100X magnifications.
(TIF)

**S21 Fig. Microscopic images of Lactophenol cotton blue stained stained *C. albicans*.** (A) *C. albicans* M-207, (B) *C. albicans* S-470 induced on coverslip in TSB medium for (a) 24h, (b) 48h, (c) 72h, (d) 96h, (e) 120h, (f) 12days at 40X & 100X magnifications.
(TIF)

**S1 Table. Zone of inhibition for solvent extractions of garlic, clove, and Indian gooseberry by well diffusion method.** The solvents used were petroleum ether, ethyl acetate, chloroform, methanol, ethanol, and butanol.
(PDF)

## Acknowledgments

The authors wish to thank Dr. Beena, Dr. Rameez Raja, and Dr. Indumathi from Microbiology Lab at Ramaiah Teaching Hospital, Bengaluru for providing us with the *C. albicans* clinical isolates.

## Author Contributions

**Conceptualization:** Bindu Sadanandan.

**Data curation:** Bindu Sadanandan, Vaniyamparambath Vijayalakshmi, Priya Ashrit, Uddagiri Venkanna Babu, Kalidas Shetty.

**Formal analysis:** Bindu Sadanandan, Uddagiri Venkanna Babu, Kalidas Shetty.

**Funding acquisition:** Bindu Sadanandan.

**Investigation:** Bindu Sadanandan, Vaniyamparambath Vijayalakshmi, Priya Ashrit, Lakavalli Mohan Sharath Kumar, Vasulingam Sampath, Amruta Purushottam Joglekar, Rashmi Awaknavar.

**Methodology:** Bindu Sadanandan, Uddagiri Venkanna Babu, Lakavalli Mohan Sharath Kumar, Vasulingam Sampath.

**Project administration:** Bindu Sadanandan.

**Resources:** Bindu Sadanandan, Uddagiri Venkanna Babu.

**Supervision:** Bindu Sadanandan, Uddagiri Venkanna Babu.

**Validation:** Bindu Sadanandan, Uddagiri Venkanna Babu.

**Writing – original draft:** Vaniyamparambath Vijayalakshmi.

**Writing – review & editing:** Bindu Sadanandan, Priya Ashrit, Uddagiri Venkanna Babu, Kalidas Shetty.

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
