## [Decision Letter · Decision Letter 0]

20 Feb 2023

PONE-D-23-01209Aqueous Spice Extracts as Alternative Antimycotics to Control Highly Drug Resistant Extensive Biofilm Forming Clinical Isolates of Candida albicansPLOS ONE

Dear Dr. Sadanandan,

Thank you for submitting your manuscript to PLOS ONE. After careful consideration, we feel that it has merit but does not fully meet PLOS ONE’s publication criteria as it currently stands. Therefore, we invite you to submit a revised version of the manuscript that addresses the points raised during the review process. Please consider the comments of both reviewers.

We look forward to receiving your revised manuscript.

Kind regards,

Guadalupe Virginia Nevárez-Moorillón, Ph.D.

Academic Editor

PLOS ONE

Journal Requirements:

4. Please upload a new copy of Figure 10 as the detail is not clear. Please follow the link for more information:

https://blogs.plos.org/plos/2019/06/looking-good-tips-for-creating-your-plos-figures-graphics/

https://blogs.plos.org/plos/2019/06/looking-good-tips-for-creating-your-plos-figures-graphics/

Reviewers' comments:

Reviewer's Responses to Questions

**Comments to the Author**

1. Is the manuscript technically sound, and do the data support the conclusions?

Reviewer #1: Yes

Reviewer #2: Yes

2. Has the statistical analysis been performed appropriately and rigorously? 

Reviewer #1: Yes

Reviewer #2: Yes

3. Have the authors made all data underlying the findings in their manuscript fully available?

Reviewer #1: Yes

Reviewer #2: Yes

4. Is the manuscript presented in an intelligible fashion and written in standard English?

Reviewer #1: Yes

Reviewer #2: Yes

5. Review Comments to the Author

Reviewer #1: The aim of this study was to explore the potential of spice-based antimycotics to control C. albicans biofilms. The methods and study design are appropriate for answering the research question and the experiments have appropriate control group. The statistical tests used were appropriate and correctly reported. The figures are clear and they accurately represent the results. Previous research by the authors and others has been discussed and those results have been compared to the current results. The results support the conclusions. Minor revisions are necessary:

1. Please review the English.

2. In the Abstract, please, specify what TSA means.

3. In the Introduction, please added more information about the cytotoxic effects of the Spice Extracts used in this study.

4. In addition, the cytotoxicity or biocompatibility of Spice Extracts, at the same concentrations used in this study, should be evaluated.

5. In the MM, please, specify what ICU means.

Reviewer #2: The paper by Bindu Sadanandan et al describes the effect of aqueous spice extracts on highly drug resistant extensive biofilm forming clinical isolates of Candida albicans. Taking into account the understanding of the impact of biofilms on bacterial susceptibility to antimicrobials, as well as the wide spread of resistance to antimicrobials, this topic is very important for the clinical practice.

Overall the experimental design is relevant and idea is publication-worth.

Nevertheless, the some issues should be addressed before considering for publication and paper requires the revision.

For regret the quality of figures is too low to evaluate them, the font size in some cases should be increased.

It looks a bit strange to use MTT at half-MIC concentration, in such design the assay is not informative. It would be nice to check whether the aqueous spice extracts are effective against preformed biofilms. Briefly, one should treat pre-formed biofilms with extracts for some time frames and then evaluate the residual viability of cells in biofilm with MTT.

It would be worse to quantify residual biomass on Figs 11 and 13, there are many free software for quantification of biofilms and biofilm-embedded cells from fluorescent microscopic images.

The coating of surfaces with aqueous spice extracts seems to be problematic, the reviewer recommends to change the focus to the treatment of Candida-associated mucosal (oral, vaginal) infections since the aqueous fraction will not irritate the mucosa. In this relation a some information about these infection should be added to Introduction.

6. PLOS authors have the option to publish the peer review history of their article (what does this mean?). If published, this will include your full peer review and any attached files.

Reviewer #1: No

Reviewer #2: No

---

## [Author Response · Author response to Decision Letter 0]

17 Apr 2023

RESPONSE TO THE REVIEWER’S COMMENTS

Journal Requirements:

Response: The manuscript has been revised to meet PLOS ONE’s style requirements. 

Response: Not applicable

Response: The manuscript does not have any blot/gel images. In all other images, originals have been provided. 

4. Please upload a new copy of Figure 10 as the detail is not clear. Please follow the link for more information: https://blogs.plos.org/plos/2019/06/looking-good-tips-for-creating-your-plos-figures-graphics/https://blogs.plos.org/plos/2019/06/looking-good-tips-for-creating-your-plos-figures-graphics/

Response: A better-quality Figure 10 has been uploaded. 

Reviewers' comments:

Reviewer's Responses to Questions

Comments to the Author

1. Is the manuscript technically sound, and do the data support the conclusions?

Reviewer #1: Yes

Reviewer #2: Yes

2. Has the statistical analysis been performed appropriately and rigorously?

Reviewer #1: Yes

Reviewer #2: Yes

3. Have the authors made all data underlying the findings in their manuscript fully available?

Reviewer #1: Yes

Reviewer #2: Yes

4. Is the manuscript presented in an intelligible fashion and written in standard English?

Reviewer #1: Yes

Reviewer #2: Yes

5. Review Comments to the Author

Reviewer #1: The aim of this study was to explore the potential of spice-based antimycotics to control C. albicans biofilms. The methods and study design are appropriate for answering the research question and the experiments have appropriate control group. The statistical tests used were appropriate and correctly reported. The figures are clear and they accurately represent the results. Previous research by the authors and others has been discussed and those results have been compared to the current results. The results support the conclusions. Minor revisions are necessary:

1. Please review the English.

Response: The English has been reviewed and the manuscript has been revised accordingly. 

2. In the Abstract, please, specify what TSA means.

Response: TSA has been specified as Tryptic Soy Agar in the abstract. 

3. In the Introduction, please added more information about the cytotoxic effects of the Spice Extracts used in this study.

Response: Several of the spice extracts are found to be non-toxic to normal cells and toxic to cancer cells. Information regarding the cytotoxic effect of the spice extracts has been included in the introduction.

4. In addition, the cytotoxicity or biocompatibility of Spice Extracts, at the same concentrations used in this study, should be evaluated.

Response: The biocompatibility of the spice extracts has also been discussed in the introduction.

5. In the MM, please, specify what ICU means.

Response: ICU has been specified as Intensive Care Unit in the materials and methods.

Reviewer #2: The paper by Bindu Sadanandan et al describes the effect of aqueous spice extracts on highly drug resistant extensive biofilm forming clinical isolates of Candida albicans. Taking into account the understanding of the impact of biofilms on bacterial susceptibility to antimicrobials, as well as the wide spread of resistance to antimicrobials, this topic is very important for the clinical practice.

Overall the experimental design is relevant and idea is publication-worth.

Nevertheless, the some issues should be addressed before considering for publication and paper requires the revision.

For regret the quality of figures is too low to evaluate them, the font size in some cases should be increased.

Response: We have done our best to improve the quality of the figures.

It looks a bit strange to use MTT at half-MIC concentration, in such design the assay is not informative. 

Response: Minimum inhibitory concentration 50 (MIC50) is the most widely used and informative measure of a drug’s efficacy. However, we have carried out MTT at MIC concentrations lower and higher than MIC50 (Fig 7). 

It would be nice to check whether the aqueous spice extracts are effective against preformed biofilms. Briefly, one should treat pre-formed biofilms with extracts for some time frames and then evaluate the residual viability of cells in biofilm with MTT.

Response: We have performed experiments to check the effectiveness of garlic extract on preformed biofilm. Biofilms were grown for 24 and 48 h and then treated with garlic extract at different concentrations and incubation times and the residual viability of cells was evaluated by MTT assay. The results have been included in the results and discussion section. 

It would be worse to quantify residual biomass on Figs 11 and 13, there are many free software for quantification of biofilms and biofilm-embedded cells from fluorescent microscopic images.

Response: The integrated intensity of the cultures (control and treated with spice extracts) were quantified using ImageJ. The results of the same have been included in the results and discussion section of the manuscript.

The coating of surfaces with aqueous spice extracts seems to be problematic, the reviewer recommends to change the focus to the treatment of Candida-associated mucosal (oral, vaginal) infections since the aqueous fraction will not irritate the mucosa. In this relation some information about these infections should be added to Introduction.

Response: In our study we are not trying to coat the surface with aqueous spice extract, rather inhibit the growth of Candida on surfaces using spice extracts. However as per the reviewer’s suggestion, as we are using aqueous extracts of spices, the information regarding Candida associated mucosal infections has been included in the introduction. 

6. PLOS authors have the option to publish the peer review history of their article (what does this mean?). If published, this will include your full peer review and any attached files.

Do you want your identity to be public for this peer review? For information about this choice, including consent withdrawal, please see our Privacy Policy.

Reviewer #1: No

Reviewer #2: No

---

## [Decision Letter · Decision Letter 1]

25 Apr 2023

PONE-D-23-01209R1Aqueous Spice Extracts as Alternative Antimycotics to Control Highly Drug Resistant Extensive Biofilm Forming Clinical Isolates of Candida albicansPLOS ONE

Dear Dr. Sadanandan,

Thank you for submitting your manuscript to PLOS ONE. After careful consideration, we feel that it has merit but does not fully meet PLOS ONE’s publication criteria as it currently stands. Therefore, we invite you to submit a revised version of the manuscript that addresses the points raised during the review process.

The reviewers have commented that their suggestions have been addressed. The revised manuscript still needs modifications, mainly in the Conclusions section, which needs to be modified to comply with PLOS One guidelines. Usually, the conclusions are concrete statements based on objectives and results. Please revise.  Please submit your revised manuscript by Jun 09 2023 11:59PM. If you will need more time than this to complete your revisions, please reply to this message or contact the journal office at plosone@plos.org. Please include the following items when submitting your revised manuscript:A rebuttal letter that responds to each point raised by the academic editor and reviewer(s). You should upload this letter as a separate file labeled 'Response to Reviewers'.A marked-up copy of your manuscript that highlights changes made to the original version. You should upload this as a separate file labeled 'Revised Manuscript with Track Changes'.An unmarked version of your revised paper without tracked changes. You should upload this as a separate file labeled 'Manuscript'.If applicable, we recommend that you deposit your laboratory protocols in protocols.io to enhance the reproducibility of your results. Protocols.io assigns your protocol its own identifier (DOI) so that it can be cited independently in the future. For instructions see: https://journals.plos.org/plosone/s/submission-guidelines#loc-laboratory-protocols. Additionally, PLOS ONE offers an option for publishing peer-reviewed Lab Protocol articles, which describe protocols hosted on protocols.io. Read more information on sharing protocols at https://plos.org/protocols?utm_medium=editorial-email&utm_source=authorletters&utm_campaign=protocols.

We look forward to receiving your revised manuscript.

Kind regards,

Guadalupe Virginia Nevárez-Moorillón, Ph.D.

Academic Editor

PLOS ONE

Journal Requirements:

Reviewers' comments:

Reviewer's Responses to Questions

**Comments to the Author**

1. If the authors have adequately addressed your comments raised in a previous round of review and you feel that this manuscript is now acceptable for publication, you may indicate that here to bypass the “Comments to the Author” section, enter your conflict of interest statement in the “Confidential to Editor” section, and submit your "Accept" recommendation.

Reviewer #1: All comments have been addressed

Reviewer #2: All comments have been addressed

2. Is the manuscript technically sound, and do the data support the conclusions?

Reviewer #1: Yes

Reviewer #2: Yes

3. Has the statistical analysis been performed appropriately and rigorously? 

Reviewer #1: Yes

Reviewer #2: Yes

4. Have the authors made all data underlying the findings in their manuscript fully available?

Reviewer #1: Yes

Reviewer #2: Yes

5. Is the manuscript presented in an intelligible fashion and written in standard English?

Reviewer #1: Yes

Reviewer #2: Yes

6. Review Comments to the Author

Reviewer #1: (No Response)

Reviewer #2: (No Response)

7. PLOS authors have the option to publish the peer review history of their article (what does this mean?). If published, this will include your full peer review and any attached files.

Reviewer #1: No

Reviewer #2: No

---

## [Author Response · Author response to Decision Letter 1]

22 May 2023

RESPONSE TO THE REVIEWER’S COMMENTS

PONE-D-23-01209R1

Aqueous Spice Extracts as Alternative Antimycotics to Control Highly Drug Resistant Extensive Biofilm Forming Clinical Isolates of Candida albicans

The reviewers have commented that their suggestions have been addressed. The revised manuscript still needs modifications, mainly in the Conclusions section, which needs to be modified to comply with PLOS One guidelines. Usually, the conclusions are concrete statements based on objectives and results. Please revise. 

RESPONSE: The conclusions section of the manuscipt has been revised according to the PLOS ONE guidelines. 

Journal Requirements:

RESPONSE: Minor revisions in the refernces have been made as indicated in the track changes of the manuscript.

Reviewers' comments:

Reviewer's Responses to Questions

Comments to the Author

1. If the authors have adequately addressed your comments raised in a previous round of review and you feel that this manuscript is now acceptable for publication, you may indicate that here to bypass the “Comments to the Author” section, enter your conflict of interest statement in the “Confidential to Editor” section, and submit your "Accept" recommendation.

Reviewer #1: All comments have been addressed

Reviewer #2: All comments have been addressed

2. Is the manuscript technically sound, and do the data support the conclusions?

Reviewer #1: Yes

Reviewer #2: Yes

3. Has the statistical analysis been performed appropriately and rigorously?

Reviewer #1: Yes

Reviewer #2: Yes

4. Have the authors made all data underlying the findings in their manuscript fully available?

Reviewer #1: Yes

Reviewer #2: Yes

5. Is the manuscript presented in an intelligible fashion and written in standard English?

Reviewer #1: Yes

Reviewer #2: Yes

6. Review Comments to the Author

Reviewer #1: (No Response)

Reviewer #2: (No Response)

7. PLOS authors have the option to publish the peer review history of their article (what does this mean?). If published, this will include your full peer review and any attached files.

Do you want your identity to be public for this peer review? For information about this choice, including consent withdrawal, please see our Privacy Policy.

Reviewer #1: No

Reviewer #2: No

---

## [Editor Report · Decision Letter 2]

1 Jun 2023

Aqueous Spice Extracts as Alternative Antimycotics to Control Highly Drug Resistant Extensive Biofilm Forming Clinical Isolates of Candida albicans

PONE-D-23-01209R2

Dear Dr. Sadanandan,

We’re pleased to inform you that your manuscript has been judged scientifically suitable for publication and will be formally accepted for publication once it meets all outstanding technical requirements.

Kind regards,

Guadalupe Virginia Nevárez-Moorillón, Ph.D.

Academic Editor

PLOS ONE
---

## [Editor Report · Acceptance letter]

5 Jun 2023

PONE-D-23-01209R2 

Aqueous Spice Extracts as Alternative Antimycotics to Control Highly Drug Resistant Extensive Biofilm Forming Clinical Isolates of *Candida albicans*

Dear Dr. Sadanandan:

I'm pleased to inform you that your manuscript has been deemed suitable for publication in PLOS ONE. Congratulations! Your manuscript is now with our production department. 

Kind regards, 

on behalf of

Dr. Guadalupe Virginia Nevárez-Moorillón 

Academic Editor

PLOS ONE